# A Scoping Review of Non-Pharmacological, Non-Surgical Secondary Prevention Strategies in Ischaemic Stroke and TIA in National Stroke Guidelines and Clinical Audit Documents

**DOI:** 10.3390/healthcare10030481

**Published:** 2022-03-04

**Authors:** Patricia Hall, Lena von Koch, Xu Wang, Olive Lennon

**Affiliations:** 1iPASTAR Collaborative Doctoral Award Programme, RCSI Division of Population Health Sciences, RCSI University of Medicine and Health Sciences, D02 H903 Dublin, Ireland; olive.lennon@ucd.ie; 2School of Public Health, Physiotherapy and Sports Science, Health Science Centre, University College Dublin, D04 C7X2 Belfield, Ireland; 3Health Services Research, Department of Neurobiology, Care Sciences and Society, Karolinska Institutet, Alfred Nobels Allé 23, 141 83 Huddinge, Sweden; lena.von.koch@ki.se; 4Psychology/Leeds School of Social Sciences, Leeds Beckett University, CL803, Leeds LS1 3HE, UK; x.wang@leedsbeckett.ac.uk

**Keywords:** stroke, secondary prevention, non-pharmacological, non-surgical, clinical guidelines, clinical audit

## Abstract

The Stroke Action Plan for Europe (2018–2030) calls for national-level secondary prevention plans that address lifestyle, in addition to prevention medications and surgical interventions. This scoping review examines national stroke care guideline and audit documents across WHO regions to identify non-pharmacological, non-surgical stroke secondary prevention recommendations and associated performance indicators. Using a snowballing methodology, 27 guideline documents met the inclusion criteria. Sixteen (59%) detailed, non-pharmacological, non-surgical stroke secondary prevention-addressing known, modifiable population attributable risk factors, of physical inactivity (N = 11), smoking (N = 11), unsafe alcohol consumption (N = 10), diet (N = 8), weight (N = 5), stress (N = 4) and depression (N = 2). Strategies recommended to address these risk factors were: assessment of stroke risk/risk factors (N = 4); provision of advice and information on reducing lifestyle related risk (N = 16); education and counselling for lifestyle behaviour change (N = 8) and onward referral for specialist management of risk (N = 4). Of the nine stroke audits/registries identified, only three (33%) included non-pharmacological, non-surgical quality indicators of documented provision of advice or information on the following: general lifestyle (N = 2); smoking cessation for current smokers (N = 2); reduction in alcohol consumption, where relevant (N = 1), exercise participation (N = 1) and diet (N = 1). Preventive quality indicators addressing the management of weight, stress or depression were absent. This review highlights current gaps in optimal stroke secondary prevention recommendations and their implementation.

## 1. Introduction

The burden of stroke is increasing, despite significant advances in acute stroke medical management [1] with disability-adjusted life years, due to stroke continuing to rise sharply [2,3]. Following transient ischaemic attack (TIA) or ischaemic stroke, the cumulative risk of a recurrent cardiovascular event at ten years is 39.2% [4], with higher associated death and disability levels [5]. Optimised secondary prevention strategies after stroke need to address all modifiable risk factors. The INTERSTOKE case control study identifies ten population attributable risk (PAR) factors that account for 90% of ischaemic stroke globally [6]. All present legitimate targets for optimising stroke secondary prevention: hypertension, physical activity, dyslipidaemia, diet, central adiposity, psychosocial factors (stress from home and/or work, life events and depression), current smoking, cardiac causes, high or heavy episodic alcohol consumption and diabetes mellitus. While cumulative healthy lifestyle behaviours are well established as important in healthy aging and primary stroke prevention [7,8], adjunctive lifestyle modification in the setting of established cardiovascular disease (CVD) is arguably of equal importance [9].

Identified gaps in comprehensive stroke care include actionable messaging for secondary prevention [10]. Important lessons can be learned in stroke from comparator populations with CVD, where, for example, just three healthy lifestyle behaviours (smoking cessation, regular exercise, and healthy eating) at 30 days post–hospital discharge from myocardial infarction demonstrate a 4-fold decreased risk of death, re-infarction, and stroke after 6 months, when compared with those who adhered to none of these behaviours [11], and where comprehensive risk reduction programmes, such as cardiac rehabilitation, have proven efficacy [12]. Combinations of healthy lifestyle factors are observed to have a dose-dependent association with lower all-cause and cardiovascular mortality for individuals with an existing stroke diagnosis [13]. Furthermore, modelling in stroke suggests that lifestyle changes in diet and exercise, in addition to optimised pharmacotherapy after stroke, could result in an 80% reduction in recurrent vascular events at 5 years [14]. 

Emphasis on stroke prevention is now clearly reflected in the Stroke Action Plan for Europe (SAP-E) 2018–2030 [15]. In secondary prevention, specifically, national-level stroke plans are called for that include follow-up in primary/community care and ensure access to key preventative strategies addressing lifestyle, in addition to secondary prevention medications (antihypertensives, lipid lowering agents, antiplatelets, anticoagulants, oral hypoglycaemic agents and insulin) and surgical interventions (carotid endarterectomy, and patent foramen ovale (PFO) closure) are further targeted. Non-pharmacological, non-surgical stroke secondary prevention, defined by an expert Delphi consensus, comprehensively addresses risk-reducing health behaviours that include adherence with secondary prevention medication prescription, physical activity participation, consuming a healthy diet, engaging in no or safe alcohol consumption levels, smoking abstinence and self-management of mood and psychological distress [16]. At the national level, targeted by SAP-E, clinical guidelines are critical to quality of care as ‘systematically developed statements to assist practitioner and patient decisions about appropriate health care for specific clinical circumstances’ [17] and can positively impact patient outcomes when implemented well [18,19]. Therefore, stroke guidelines that address secondary prevention play an important role in optimising the widespread adoption of all beneficial interventions for risk reduction. Critical to the success of such clinical guidelines are the strategies and methods used for their implementation, often driven and documented by registry or audit activities [20]. The association between adherence to recommended guidelines in both acute stroke and rehabilitation are well documented, with respect to health and recovery outcomes [21,22]. Similarly, the ‘Get with the Guidelines-Stroke’ process improvement programme in the US identified significant and clinically meaningful improvements in 11 out of 13 measures in stroke care [23]. However, only two of these targets addressed lifestyle-related PAR factors for stroke, those of smoking and weight reduction counselling.

Given the pressing need to maximise stroke secondary prevention strategies and the importance of national guidelines and programmes, the purpose of this paper is to identify the recommendations and gaps for non-pharmacological, non-surgical stroke secondary prevention [16] in published stroke guideline documents, and map them to PAR factors in ischaemic stroke, where lifestyle/behavioural interventions are the first line preventive treatment [6]. The second aim of this study is to identify the key performance indicators used by national clinical audits or registries that capture adjunctive, lifestyle interventions in quality reviews of stroke care on a national basis. 

## 2. Methods

A scoping review of contemporary stroke guideline documents and published national stroke audits was undertaken. A scoping review was the methodology of choice as the purpose of this study was to identify and map the concept of non-pharmacological non-surgical stroke secondary prevention strategies in national guidelines and audits of stroke care across multiple jurisdictions. This proposed strategy allows examination of current best practice recommendations and their potential implications for policy makers, practitioners and consumers in stroke care who may lack resources to undertake the work themselves [24,25]. The review conforms to the Arskey and O’Malley framework for conducting scoping reviews [24] which recognises the iterative nature of the process and variety of mechanisms required for searching for relevant information to achieve broad results. From the outset, to be as comprehensive as possible, a strategy involving an existing international network of stroke secondary prevention researchers (INSsPiRE) was used to identify relevant national documents. The conduct of the scoping review was guided by the Joanna Briggs Institute published methodological guidance for the conduct of scoping reviews [26] and consistent with this guidance no formal quality assessment of the documents was performed. 

### 2.1. Stage 1: Identifying the Research Question

The recommended Population, Concept, and Context (PCC) mnemonic [27] for scoping reviews was used to guide the research question. The population of interest was identified as individuals post-ischaemic stroke or TIA. Potential secondary prevention targets under consideration were derived from the INTERSTROKE study [6] modifiable PAR factors for ischaemic stroke, as summarised in Table 1, and aligned to the published definition of non-pharmacological, non-surgical stroke secondary prevention [16]. The factors chosen exclude PAR factors where pharmacological agents and/or surgery would play a primary role in management (e.g., diabetes, hypertension, cardiac causes). The context of the review relates to all healthcare sectors where stroke secondary prevention may be addressed and includes acute, rehabilitation and primary/community care. 

#### Review Objective

(1) To identify and map the current recommendations for stroke secondary prevention in national stroke guideline documents to attributable, modifiable risk factors in ischaemic stroke, where pharmacological or surgical interventions are not the first line preventive treatment; (2) to identify where the current emphasis on non-pharmacological, non-surgical stroke secondary prevention is in clinical practice by examining the key performance indicators used in national clinical audits or registries of stroke care. Specifically, the scoping review asks the following: 

What lifestyle/behavioural (non-pharmacological non-surgical) recommendations for stroke secondary prevention exist in national or multiple jurisdiction stroke guidelines?

What key performance indicators are used for non-pharmacological, non-surgical stroke secondary prevention in practice in national clinical audits or registries of national stroke services?

### 2.2. Stage 2: Identifying Relevant Sources

In keeping with the guidance on scoping reviews [24] to comprehensively identify relevant documents, a snowballing approach was employed whereby members of the INSsPiRE network were asked to identify through the grey literature:The most recent Clinical Stroke Guideline document developed by their national stroke organisation or governmental health department and available on the organisation’s website or in the published literature. Where a specific stroke secondary prevention guideline was published and available, it was to be identified in preference to a general stroke guideline document. Otherwise a general stroke management guideline was to be identified and the section addressing secondary prevention was to be reviewed.National audits of stroke care or, in the absence of a national audit, national stroke registry data that had defined quality criteria.

For any document identified, the most up-to-date version available on the organisation’s website or in the published literature was to be selected.

Based on the geographical spread provided by this method, INSsPiRE members were then asked to identify further experts in stroke secondary prevention from their wider networks to cover additional jurisdictions that were not included by the initial search. Adequate representation from the continents of the Americas, Australasia, Asia, Europe and Africa were sought through this method.

The targeted search was initiated in June 2020 and terminated in November 2020.

#### Inclusion Criteria

To be included in this scoping review, guideline documents returned must have been at national or continental level and published by a national or international stroke organisation or governmental health department and relate directly to stroke care. Similarly, clinical audits of stroke care or stroke registry data with quality criteria were required to have been conducted at a national level and conducted by a nationally recognised body. Each guideline and audit document returned was reviewed independently against these inclusion criteria by two reviewers.

### 2.3. Stage 3: Extracting and Charting the Results

The ‘descriptive analytical’ method within the narrative tradition was employed, applying a common analytical framework to all the primary reports included and collection of standardised information from each report was applied. Guideline documents and audits in languages other than English were read and translated by INSsPiRE network colleagues proficient in the given language. 

Data were charted using a proforma in Microsoft Excel. We recorded information as follows from guideline documents:Author(s), year of publication, jurisdiction covered, whether the guideline was a general stroke care guideline or specific to stroke secondary preventionNon-pharmacological, non-surgical recommendation/s made listed under the heading of stroke secondary preventionThe approaches recommended for non-pharmacological, non-surgical stroke secondary prevention

For audits of stroke care, we extracted the following information:Author(s), year of publication and jurisdiction covered.Key performance indicators used for assessing current clinical practice in relation to non-pharmacological, non-surgical stroke secondary prevention strategies.Rating of achievement of the key performance indicators.

Data extracted to our proforma from the multiple documents identified were collated and presented in the following ways. Firstly, by summary statistics accompanied by a narrative synthesis of stroke guidelines addressing recommendations in the area of non-pharmacological, non-surgical stroke secondary prevention; secondly, by summarising the available clinical audits in stroke care and the key performance indicators used in non-pharmacological, non-surgical stroke secondary prevention. 

## 3. Results

In total, thirty guideline documents were returned from the snowballing methodology employed. The guideline documents from Kenya, Uganda and Indonesia were excluded as they were more general healthcare guidelines that did not relate specifically to stroke. Guideline documents and clinical audits returned that were eligible for inclusion are summarised (N = 27) in Table 2, under the World Stroke Organisation (WSO) regional groupings of Sub-Saharan Africa/Middle East/East Mediterranean, Americas, Europe and Asia/Oceania. One WSO Global Guideline document [28] was included along with two additional guideline documents from regional areas (Europe [15], and Australia/New Zealand [29]). The remainder represent twenty-four individual countries. These documents represented a mix of generic stroke management guidelines and specific stroke prevention guidelines.

Of the twenty-seven guideline documents identified, sixteen addressed non-pharmacological, non-surgical stroke secondary prevention recommendations; six of these were specific stroke secondary prevention guideline documents. The remaining ten were general stroke care guidelines. Most included a specific section detailing stroke secondary prevention recommendations that included non-pharmacological, non-surgical stroke secondary prevention. However, one of these referred to other guideline documents, relating to cardiovascular disease prevention in general, but not stroke specific and not secondary prevention.

Table 3 presents the modifiable risk factors cited and the approaches recommended to address these in the guideline documents identified. Where documented, recommendations in relation to non-pharmacological, non-surgical risk factor management after stroke followed four principal strategic approaches, as outlined in Table 3. (1) Assessment of stroke risk/risk factors; (2) provision of advice and information on reducing lifestyle related risk; (3) education and counselling for lifestyle behaviour change and (4) onward referral for specialist management of risk. Two documents that addressed non-pharmacological, non-surgical stroke secondary prevention avenues recommended that, generally, lifestyle risk factors should be identified and addressed in secondary prevention, without additional detail or identifying specific risk factors.

A total of nine national stroke care audits or registry-based surveys with quality indicators were identified, covering Canada, Australia/New Zealand, Denmark, Ireland, Netherlands, Norway, Scotland, Sweden, UK. Three reported quality indicators that addressed topic/s related to non-pharmacological, non-surgical stroke secondary prevention are presented in Table 4. The performance indicators utilised included: whether provision of general risk factor/lifestyle behaviour change advice or information prior to discharge was documented; the number of smokers documented as given specific advice to stop smoking; whether reduction in alcohol consumption was advised where relevant and whether discussion on exercise participation and healthy diet took place.

## 4. Discussion

The scoping review presented in this paper was conducted as a staged body of work, adhering to a published framework [24,65], and is the first that examines the key non-pharmacological, non-surgical stroke secondary prevention recommendations being made to guide clinical practice internationally. While not an exhaustive compendium of stroke guideline documents globally, broad representation from WHO regional groups was achieved and major regional and census-driven documents, such as the European Stroke Organization (ESO) and American Heart Association (AHA)/American Stroke Association (ASA), are presented. As a scoping review, we did not follow the Equator Network reporting recommendations here, nor use their Appraisal, Research and Evaluation (AGREE II) instrument [66], as no appraisal of the quality of the guideline documents identified was formally conducted. Previous research has indicated that not all stroke guideline documents are developed with the same methodological rigour [67,68]. However, the quality has been noted to have improved systematically in more recent years [69,70]. This must be considered in interpreting the findings of the review. The authors, furthermore, acknowledge that lifestyle factors are contextual to different income countries and cultures. While efforts were made in our snowballing method to extend our search to low and low-middle income countries, the search yielded a low response in these categories. One low-income country (GNI per capita < USD 1046) and two low-middle income countries (GNI per capital USD 1046–4095), as defined by the World Bank (2020–2021) [71], were accessed using our scoping methodology. Generic guideline documents provided by these countries were subsequently excluded based on the a priori review criteria. No stroke care audit documents were subsequently reported for low or low-middle income countries.

A key finding from this scoping review is that non-pharmacological, non-surgical interventions for stroke secondary prevention are not a current mainstay in guideline documents for stroke care and secondary prevention. Recommendations relating to risk reducing health behaviours were made in just over half of the guideline documents identified, indicating a lack of direction for clinicians on the important role of lifestyle-related risk factor reduction in stroke secondary prevention and the need to target and address all identified and modifiable PAR factors for stroke. Where lifestyle-related recommendations were present in the guideline documents identified, there was no consistency in the message delivered to clinicians across the documents. While the focus was primarily centred on smoking cessation, physical activity participation, healthy eating and safe alcohol consumption, as risk reducing strategies after stroke, important avenues for secondary prevention, including management of depression and psychological distress and weight management, were less frequently addressed and were never considered as secondary prevention performance indicators in the clinical audits conducted. Depression after stroke has been identified in a recent meta-analysis as an independent predictor of stroke recurrence in ischaemic stroke patients [72]. Similarly, stressful life events and poor adaptation to stress are independently associated with increased risk of stroke [73,74,75,76], and psychological distress has been identified as a predictor of fatal ischaemic stroke [77]. While an obesity paradox is reported in stroke, with respect to vascular events and mortality, recent evidence has identified it as a significant risk factor for recurrent stroke [78,79,80].

SAP-E 2018–2030 addresses challenges facing stroke survivors and families, associated with life after stroke, including the assessment and management of modifiable risk factors in the prevention of recurrent stroke [15]. Individuals after stroke are known to have low adherence with lifestyle and cumulative lifestyle-related health behaviours [81] and show significantly lower engagement in health behaviours in comparison to matched controls [82]. Only one-third of the guideline documents reviewed in this study recommended routinely assessing for lifestyle-related risk factors. Yet all these documents recommended providing advice and information. This mismatch suggests tailored and individualised information provision after stroke, in relation to secondary prevention, is not being recommended as a gold standard. However, while active information provision after stroke (one that establishes what information needs an individual has and adapts the provision to make it relevant and usable) improves stroke-survivors’ knowledge and quality of life and may help reduce anxiety and depression [83]; both knowledge and attitude change are considered necessary precursors of behaviour change [84]. Poor health behaviours are well entrenched and difficult to change, and it is well documented that simply giving people information and advice does not change behaviour [84,85,86,87]. Half of all guideline documents that addressed non-pharmacological, non-surgical stroke secondary prevention recommended education and counselling to address behavioural risk factors, without providing additional detail. Successful behaviour change is difficult, requires sustained motivation and support, is located in complex social environments and cultures and should be underpinned with evidence-based behaviour change theory and strategies [84].

The next key finding from this review relates to clinical audit. Overall, the promotion of guideline recommendations addressing modifiable health behaviours in stroke secondary prevention in routine practice, by means of clinical audit performance indicators, was lacking. Where present, performance indicators measured whether general lifestyle and/or smoking cessation advice was provided, rather than the behaviour change achieved. As a model, this tends to work less well in more chronic conditions, such as stroke, and is even less effective in the sphere of changing behaviour [88]. Only one document in this study audited physical activity, dietary or safe alcohol consumption promotion in stroke secondary prevention. Weight management and addressing psychosocial stress or depression did not feature in any of the documents in this review. These findings fall in line with a European survey of national stroke societies and stroke experts, where the general provision of stroke secondary prevention services was reported as varied, inconsistent and largely dependent on national income [89]. Review of the quality of stroke care was, therefore, often lacking and likely associated with the gaps in the provision of the services identified [90].

Both the guideline documents and the audit performance indicators identified in this review reflect a focus internationally on the acute management of stroke. Post-acute stroke care has previously been identified as variable and fragmented [91], and with limited or no quality appraisal [10]. Indeed, a paradigm shift has been called for in comprehensive stroke care and outcomes, including rehabilitation and public health for secondary prevention [10]. Since 2019, the updated American Stroke Association Recommendations for the Establishment of Stroke Systems of Care now explicitly states that stroke centres should adopt approaches to secondary prevention that address all major modifiable risk factors, for all patients with a history or a suspected history of stroke or transient ischaemic attack, and the focus of post-acute care should be on reducing mortality, maximizing recovery, and preventing recurrent stroke and cardiovascular events. They further recommend that community services reinforce the secondary prevention and self-management of stroke risk factors and lifestyle changes to decrease the risk of recurrent stroke with trained stroke nurses, nurse practitioners, social workers, community health workers, and others playing a pivotal role [92].

## 5. Limitations

A number of limitations with respect to this scoping review are acknowledged. The quantity of international documents presented is not exhaustive and a decision was made to terminate the snowballing method for guideline identification in November 2020, when a sample deemed sufficiently representative of geographical areas was identified. We acknowledge that other stroke guideline documents exist and that some reported here have been updated since this scoping review was conducted (e.g., live guideline documents and others). Indeed, both the AHA/ASA guidelines and the Australian living guidelines have been significantly revised since this scoping review was completed [29,34]. Both documents now signal an increasing awareness of the value of non-pharmacological, non-surgical stroke secondary prevention avenues and make specific evidence-based recommendations about smoking cessation, diet, physical activity, substance use (including alcohol), obesity [29,93], multi-modal behavioural interventions after stroke [93] and adherence to stroke secondary prevention medication as a lifestyle behaviour [29].

Our strategy to target risk reduction recommendations that were detailed under a secondary prevention heading in the identified documents means it is possible that issues, such as management of post-stroke depression, for example, may be addressed elsewhere in stroke guideline documents. While these recommendations in the management of depression may have the potential to impact stroke recurrence, we opted to only extract recommendations that were made specifically for targeted stroke secondary prevention and acknowledge that overlap between other recommendations may exist.

## 6. Conclusions

This scoping review highlights, in the clinical guideline documents reviewed, a current lack of emphasis on non-pharmacological, non-surgical stroke secondary prevention recommendations addressing legitimate modifiable risk factors after stroke. This finding is further reflected in an absence of quality indicators relating to non-pharmacological, non-surgical stroke secondary prevention in national stroke care audits. These gaps identified provide a basis for future development in this area.

## Figures and Tables

**Table 1 healthcare-10-00481-t001:** Population Attributable Risk Factors for Ischaemic Stroke from the INTERSTROKE study and as included in the current Review.

Modifiable Risk Factors (Interstroke Study)	Population Attributable Risk (PAR) %	99% CI	Considered in Current Review
Hypertension	34.6	30.4–39.1	X
Physical activity	28.5	14.5–48.5	√
Diet	18.8	11.2–29.7	√
Central adiposity	26.5	18.8–36.0	√
Smoking	12.4	10.2–14.4	√
Excessive alcohol consumption	3.8	0.9–14.4	√
Psychosocial stress from home/work/life events	4.6	2.1–9.6	√
Depression	5.2	2.1–9.8	√
Diabetes	5.0	2.6–9.5	X
Dyslipidaemia	26.8	22.2–31.9	X

**Table 2 healthcare-10-00481-t002:** Stroke Guideline Documents and Audits included in the Review by WSO Regional Grouping.

World Stroke Organisation (WSO) Regional Groups	Guideline Documents Returned	Guidelines–Individual Countries	Audit/Survey Registry Data	Country
Sub-Saharan Africa/Middle East/Eastern Mediterranean	2	South Africa [30,31]; Turkey [32,33]	0	
Americas	8	USA *^$^ [34]; Canada ^$^ [35]; Argentina [36]; Brazil [37]; Chile [38]; Colombia [39]; Mexico ^$^ [40]; Peru [41];	1	Canada [42]
Europe	13	European Stroke Organisation *^$^ [15]; Austria [43]; Denmark ^$^ [44]; Germany [45]; Ireland ^$^ [46]; Italy * [47]; Netherlands ^$^ [48]; Norway *^$^ [49]; Poland ^$^ [50]; Scotland ^$^ [51]; Spain *^$^ [52]; Sweden ^$^ [53]; UK ^$^ [54]	7	Denmark [55]; Ireland [56]; Netherlands [57]; Norway [58]; Scotland [59]; Sweden [60]; UK [61]
Asia/Oceania	3	Australia (inc New Zealand) ^$^ [29]; China *^$^ [62]; Japan [63]	1	Australia [64]
World Stroke Organisation	1	Global Guidelines and Action Plan ^$^ [28]	0	

As identified by reviewers: * signifies specific stroke secondary prevention guidelines; ^$^ signifies a document where non-pharmacological non-surgical stroke secondary prevention was addressed in the guidelines.

**Table 3 healthcare-10-00481-t003:** Non-pharmacological, non-surgical stroke secondary prevention modifiable risk factors and approaches recommended in guideline documents.

Modifiable Risk Factor	N = 16 Guideline Documents N
General lifestyle	11
Physical activity	11
Diet	8
Weight	5
Smoking	11
Alcohol	10
Stress	4
Depression	2
**Approach recommended for risk-reduction**	
Assessment of risk	4
Advice & information	16
Education & counselling	8
Referral to specialist	4

**Table 4 healthcare-10-00481-t004:** Stroke care audits/surveys included in the review.

Audits and Survey Registries Screened	N = 9
Addressing non-pharmacological stroke secondary prevention	3
**Modifiable population attributable secondary prevention strategies addressed**	
General lifestyle	2
Physical activity	1
Diet	1
Weight	0
Smoking	2
Alcohol	1
Stress	0
Depression	0

## Data Availability

Not applicable.

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
