# Peer review of "A Scoping Review of Non-Pharmacological, Non-Surgical Secondary Prevention Strategies in Ischaemic Stroke and TIA in National Stroke Guidelines and Clinical Audit Documents"

_healthcare, 2022, doi:10.3390/healthcare10030481_

Round 1

Reviewer 1 Report

The author prepared a concise yet comprehensive document with a focus on non-pharmacological, non-surgical secondary prevention strategies in ischaemic stroke and TIA. The manuscript could be accepted with minor revisions.

  1. It would be good if the authors give precise definition of secondary prevention strategies in ischaemic stroke and TIA as the audience of this journal is not prsented exclusively by neurologists.
  2. The authors should consider changing the title of the review. In the present form the title looks too ambiotious as gives a wrong impression to the readers that the authors identified and considered all guidelines on secondary prevention in stroke and TIA. In fact, the authors missed few important publications in the field and included some guidelines of doubtful quality. As an alternative, the authors might use one of the scales for the evaluation of clinical practice guidelines and include only quidelines of reasonable quality.

Author Response

Thank you for your positive comments and detailed feedback of our paper.

Response 1: I have expanded on what is considered to constitute stroke secondary prevention strategies in an endeavour to broaden the understanding of this concept for the wider audience. This is now available on page 1-2, lines 41 – 92 in the revised manuscript.

Response 2:To avoid any wrong impression of totality of guidelines reviewed we thank you for this observation and have changed the title accordingly. We recognise the limitations of the review methodology. The main aim of our manuscript is to explore the presence of strategies regarding non-pharmacological non-surgical stroke secondary prevention. Whilst it is not an exhaustive review of all guidelines, within the scope and timelines, we strove to obtain a representative sample. We acknowledge the reported guidelines are not all of good quality and detail this in the discussion/limitations section. It is beyond the remit of a scoping review to conduct a quality evaluation of individual documents and we identify this more clearly now in the methods section, lines 325 – 333 in the revised manuscript and in the discussion section.

Reviewer 2 Report

Authors have performed a scoping review on the non-pharmacological and non-surgical secondary prevention strategies in ischaemic stroke and TIA. The review is  important as it will inform the clinicians and researchers to focus on the factors to prevent the secondary stroke and to help people and patients with stroke to be involved in the implementation of appropriate secondary prevention strategies. Authors have acknowledged the limitations. There are comments as provided below:

1- Abstract should clearly provide, base on the current review, what strategies are provided for preventing secondary stroke.

2- Introduction could be shortened to focus on the main question and the gaps in this area.

3- Introduction, line 80, please define the PFO.

4- Methods, authors should clarify on the guideline on the scoping review as well as the snowball methodology used in the review. Why do the authors adopt both the snowball methodology as well as the guideline on the review?

5- Discussion, lines 334-337 need clarification.

6- Discussion, page 10, "Conclusions" section is missing. 

Author Response

I wish to thank you for the positive comments provided in your detailed feedback

Response 1:     In the abstract I have reworded to make more explicit the strategies provided as identified in this review. Thank you for your suggestion.

Response 2:     We have reflected and discussed widely on attempting to shorten the introduction and are keen to comprehensively situate the review in the current landscape of stroke care, in particular in relation to the gaps and challenges in relation to lifestyle related risk factors. However we very much appreciate your feedback and have notably shortened the text without losing this context and hope this is to your satisfaction.

Response 3:     PFO definition was accidentally omitted – patent foramen ovale now inserted line 112 in the revised manuscript

Response 4:     To achieve a comprehensive and representative sample of national guidelines as possible we adopted a strategy that involved different sources to identify local documents. It is permissible within the remit of a scoping review to use expert sources and networks as we have done using the snowballing method. Clarification for the reader on the use of the guidance and exploitation of networks is added on page 3 lines 325 - 333, and page 4 lines 365 - 367 in the revised manuscript

Response 5:     Clarification of lines 334-337 in the original manuscript is now included on page 9 lines 568 - 569 in the revised manuscript

Response 6:     Apologies for this omission in the original manuscript. The conclusion section is now inserted page10 lines 633 - 640

Reviewer 3 Report

I read with great attention and interest the review entitled "Non-pharmacological, non-surgical secondary prevention strategies in ischaemic stroke and TIA: a scoping review of national 3 stroke guidelines and stroke clinical audit documents. "

I believe that this review on a so often "forgotten" topic is useful and interesting.

However, I have some concerns regarding the extensive use of numbered lists in methods, and about the discussion section’s length.

Overall, I recommend this article for publication.

Author Response

Response:  

I greatly appreciate the positive comments provided and the support for this paper and thank you for your review.

I have reformatted the numbered lists in the methods section.

We very much appreciate your concerns about the length of the discussion section. We are keen to comprehensively discuss the findings in the context of the burden of stroke and the recent focus on prevention. We have revised the text to shorten this section and hope it is to your satisfaction.